# Mental Stress and Cardiovascular Health—Part I

**DOI:** 10.3390/jcm11123353

**Published:** 2022-06-10

**Authors:** Federico Vancheri, Giovanni Longo, Edoardo Vancheri, Michael Y. Henein

**Affiliations:** 1Department of Internal Medicine, S. Elia Hospital, 93100 Caltanissetta, Italy; 2Cardiovascular and Interventional Department, S. Elia Hospital, 93100 Caltanissetta, Italy; giova.longo@gmail.com; 3Department of Medical, Surgical and Advanced Technologies, Section of Neurosciences, University of Catania, 95125 Catania, Italy; edoardovancheri@hotmail.it; 4Institute of Public Health and Clinical Medicine, Umea University, 90187 Umea, Sweden; michael.henein@umu.se; 5Brunel University, Middlesex, Uxbridge UB8 3PH, UK; 6St George’s University London, London SW17 0RE, UK

**Keywords:** mental stress, coronary artery disease, mental stress-induced myocardial ischemia, heart-brain interactions, neurocardiology

## Abstract

Epidemiological studies have shown that a substantial proportion of acute coronary events occur in individuals who lack the traditional high-risk cardiovascular (CV) profile. Mental stress is an emerging risk and prognostic factor for coronary artery disease and stroke, independently of conventional risk factors. It is associated with an increased rate of CV events. Acute mental stress may develop as a result of anger, fear, or job strain, as well as consequence of earthquakes or hurricanes. Chronic stress may develop as a result of long-term or repetitive stress exposure, such as job-related stress, low socioeconomic status, financial problems, depression, and type A and type D personality. While the response to acute mental stress may result in acute coronary events, the relationship of chronic stress with increased risk of coronary artery disease (CAD) is mainly due to acceleration of atherosclerosis. Emotionally stressful stimuli are processed by a network of cortical and subcortical brain regions, including the prefrontal cortex, insula, amygdala, hypothalamus, and hippocampus. This system is involved in the interpretation of relevance of environmental stimuli, according to individual’s memory, past experience, and current context. The brain transduces the cognitive process of emotional stimuli into hemodynamic, neuroendocrine, and immune changes, called fight or flight response, through the autonomic nervous system and the hypothalamic–pituitary–adrenal axis. These changes may induce transient myocardial ischemia, defined as mental stress-induced myocardial ischemia (MSIMI) in patients with and without significant coronary obstruction. The clinical consequences may be angina, myocardial infarction, arrhythmias, and left ventricular dysfunction. Although MSIMI is associated with a substantial increase in CV mortality, it is usually underestimated because it arises without pain in most cases. MSIMI occurs at lower levels of cardiac work than exercise-induced ischemia, suggesting that the impairment of myocardial blood flow is mainly due to paradoxical coronary vasoconstriction and microvascular dysfunction.

## 1. Introduction

Targeted strategies in cardiovascular disease (CVD) to improve conventional modifiable risk factors such as hypertension, diabetes, smoking, hypercholesterolemia, and physical inactivity have led to a reduction in age-standardized cardiovascular mortality over the last five decades [1,2,3]. However, approximately one-fourth of ST-elevation myocardial infarctions occur in the absence of conventional cardiovascular risk factor, despite a higher risk of mortality compared to patients with at least one risk factor [4,5]. Moreover, several studies have identified some possible nutritional, environmental, and psychosocial risk factors, different from conventional risk factors, such as poor fetal growth, short adult height, air pollution, transportation noise, gut microbiome, and socioeconomic status that may impact disease process [6,7,8,9,10,11,12]. In recent years, mental stress has been increasingly recognized as a potentially modifiable risk factor for the development, progression, and triggering of CVD, including coronary artery disease (CAD) and stroke, independently of conventional risk factors [13,14,15,16,17,18]. Although emotional stress is a common experience in daily life, only some people experience pathological cardiovascular (CV) consequences, particularly those with significant CVD risks.

Mental stress is the negative cognitive and emotional body response to environmental demands that exceed an individual’s ability to cope [19]. Unlike conventional risk factors, definition, measurement, standardization, and reproducibility of the mental stress are complex because the concept of stress includes the interaction of external stressors and the individual’s psychological resources to cope with them [20,21]. Several mental demanding and emotionally provocative tests, such as performance of mental arithmetic tasks with time pressure or public speaking, have been used in the studies [22]. Stressors are real or perceived threats to an individual’s homeostasis, which are evaluated and appraised by the brain for their importance according to previous experience and current life.

As a result of mental stress, the brain activates physiologic hemodynamic, neuroendocrine, and immune changes through the autonomic nervous system (ANS) and the hypothalamic–pituitary–adrenal (HPA) axis. In most cases, the physiological changes induced by stress are adaptive and provide individuals with the support to cope with the changes in the environment, a condition known as “defense reaction” to prepare the individual for fight or flight [23]. Hence, the risk of a CV event induced by mental stress depends on a combination of external events and an individual’s threshold for experiencing stress, based upon genetics and early and adult life experiences [24]. This accounts for the large differences in the way people respond to stressors. Furthermore, when the association between psychological stress and CV disease is based on self-reported stress and health outcomes, there may be a tendency towards higher reporting of both stress and symptoms of disease, resulting in a spurious association [25]. This review summarizes current knowledge about the role of neural, hemodynamic, and vascular response to mental stress in the development of CV disease.

## 2. Mental Stress-Induced Myocardial Ischemia (MSIMI)

The effects of stress on cardiac function are expressed by a transient myocardial ischemic response, analogous to exercise stress-induced ischemia, in which the stress is psychological rather than physical [26]. MSIMI is a potential trigger for angina, myocardial infarction, arrhythmias, and left ventricular dysfunction [24,27,28,29]. The relationship between mental stress and MSIMI varies according to the individual’s CV risk level but is unrelated to severity of coronary obstruction or previous revascularization [30]. In the general population, free of clinically evident CAD, the hazard ratios of coronary and cerebrovascular events attributable to stress are lower compared with classic risk factors [31,32,33]. The prevalence of MSIMI in patients with stable CAD varies widely, between 20% and 70%, as the result of differences in the type of mental stress used, diagnostic methods employed, and psychological profile [34]. In patients with CAD, the presence of MSIMI is associated with a two-fold increase in the incidence of adverse cardiac events, independent of physical or pharmacological stress-induced ischemia, even after adjusting for established risk factors, such as diabetes, obesity, and sedentary life [35,36,37]. Moreover, incorporating psychosocial risk factors into the Framingham risk score improves the prediction of CAD [38]. In up to 50% of patients with stable CAD, ischemic responses are triggered not only by extremely severe emotional stress, but also by environmental challenges that may be encountered in everyday life [26,34,39]. In these patients, MSIMI occurs more frequently than exercise-induced myocardial ischemia and may also be present in those with no signs of myocardial ischemia during exercise. Indeed, while two-thirds of CAD patients showing exercise-induced ischemia have MSIMI, only half of them have exertional ischemia [40].

MSIMI is usually under-recognized because it typically occurs without pain [41,42,43]. Early Holter monitoring studies have shown that over two-thirds of myocardial ischemic episodes in CAD patients occur during daily activities, without anginal symptoms, and at lower physical exertion than observed with exercise-induced ischemia [44,45]. Additionally, MSIMI has prognostic relevance and is associated with a three-fold relative increase in mortality over 5 years in patients with CAD and positive exercise stress-test [46].

Mental stress can be acute or chronic, depending on the length and severity of exposure. Both conditions share the same CV pathological mechanisms, with only slight differences in the timing of their development. Acute mental stress is short-term exposure to severe stressors. It is more common and more frequently associated with CV events than the chronic form [47]. Acute mental stress may develop in everyday life as a result of anger, fear, acute job-strain, or stressful sport events, as well as relating to natural disasters, earthquakes, and hurricanes, or unnatural events including industrial accidents and terrorist attacks [48,49,50,51,52,53,54]. Moreover, significant increases of acute CV events have been observed in the first workday of the week and around Christmas, New Year, Easter, and the midsummer holiday [55,56,57]. Significant acute stress may trigger angina, myocardial infarction, arrhythmias, stress cardiomyopathy (Takotsubo syndrome), stroke, or sudden death [24,58,59,60]. The risk of acute coronary syndromes has been estimated to be five times higher in the 2 h after an anger outburst [50].

Chronic mental stress refers to long-term, repetitive stress exposure. Depression, anxiety, low self-esteem, loneliness, job-related stress, retirement, low socioeconomic status, and type A personality (e.g., competitive, aggressive, hostile) and type D (e.g., distressed, characterized by introversion and pessimistic emotions), are strongly linked to chronic stress, independent of established risk factors [23,61,62,63,64,65,66,67,68,69,70,71,72,73,74,75,76,77]. Moreover, acute CV events increase the risk of chronic psychological stress, such as post-traumatic disorder, which in turn may result in recurrent CV disease [78,79]. Unlike the response to acute mental stress, the association of chronic stress with increased risk of CVD is mainly due to acceleration of atherosclerosis through maladaptation of the neuroendocrine pathways involved in the response [46,58,80]. Depression is also associated with poor prognosis after myocardial infarction [81]. Meta-analyses have shown that, on average, work-related chronic stress is associated with 50% excess risk of CVD [24]. Moreover, long-term exposure to mental stress has been linked to the development of diabetes and obesity, which greatly increase CVD risk [82,83].

The impact of mental stress on CAD is part of a more general relationship between emotional factors and several chronic and acute illnesses, such as viral infections, autoimmune diseases, gastrointestinal disorder, obesity, diabetes, and cancer [84,85]. Conversely, illnesses, especially heart disease, result in negative emotion which may affect the disease progression and outcome. In addition, myocardial infarction and heart failure activate inflammatory pathways that can mediate neural changes, including neurodegeneration, which may result in long-standing cognitive dysfunction, acceleration of atherosclerosis and further triggering of CAD [86,87].

In contrast to the clinical evidence linking negative emotions to CAD, positive emotions are thought to decrease the CV reactivity to stress, heart rate, blood pressure, development of arrhythmias, thrombogenesis, and the risk of developing CAD [88,89,90,91].

## 3. Neural Control of CV Response to Mental Stress

The trigger for MSIMI is cognitive, thereby different from the physical stimuli that provoke exercise-related ischemia. Positron emission tomography (PET) imaging studies on brain correlates of mental stress induced by arithmetic task have shown that patients who develop MSIMI have a distinct increased activation of brain regions that involve memory, emotion, and sympathetic activation, compared to those with exercise-related ischemia [92]. The relationship between mental stress and MSIMI originates in the brain as appraisal of threat, based on prior experience and expectations from the current context. The appraisal is controlled by the central nervous system (CNS), which transduces the cognitive process of emotional and stressful stimuli into brain-to-body visceromotor outputs through the stress effector systems [93]. The visceromotor outputs include an increase in the activity of sympathetic nervous system, suppression of parasympathetic cardiac control, activation of HPA axis, and increase in systemic inflammation. Activation of these systems results in hemodynamic changes, alterations in coronary vasoreactivity, platelet activation, and endothelial injury, leading to myocardial ischemia, infarction, or arrhythmia. In turn, the peripheral pathophysiological changes result in body-to-brain autonomic, hormonal, and immune feedback which modulates the brain appraisal system and subjective emotional status, leading to functional and structural changes within the brain [94,95] (Figure 1).

## 4. Brain Appraisal Systems

These systems evaluate and integrate events, previous experience, and memory for their relevance to the individual. Functional neuroimaging studies using PET and functional magnetic resonance (fMRI) during experimental psychological stress have shown that the CNS response to negative emotions includes different levels of brain activation: changes within the local activity of brain regions, changes across regions, and a network of anatomically and functionally interconnected areas according to the context and the individual perception of stress [96,97,98,99,100,101].

This brain network of connections includes brainstem and subcortical regions, such as the amygdala, hypothalamus, hippocampus, periaqueductal grey (PAG), and thalamus, as well as the prefrontal cortex, especially the medial prefrontal cortex (mPFC) and anterior cingulate cortex (ACC), and insula (insular cortex) [102]. Subcortical structures, such as the amygdala, hypothalamus, and hippocampus, are currently referred to as the limbic system where the appraisal of emotion largely occurs. This system is strictly connected with the prefrontal cortex, into a cortico–limbic functional circuit, associated with initial awareness of psychological stressors and autonomic cardiovascular regulation [103,104]. Emotions are initiated, shaped, and stored in these cortical and subcortical brain regions. Environmental stimuli are processed by the thalamus which has functional connections with both cortical and limbic regions [105].

***The amygdala*** is a major brain structure mediating the interactions between stress and CV diseases. it is involved in the interpretation of relevance of environmental stimuli, integrating emotional information from other brain regions. It has a critical role in linking fearful stimuli with appropriate responses. Amygdala extracts long-term memory from past experience through projections to the hippocampus and ACC, thus modulating emotions and evaluating the predictive significance of environmental stimuli, for appropriate response to stress. Its efferent projections to cortico–limbic areas regulate the activity of the autonomic nervous system and HPA axis [106,107]. Amygdala is under tonic inhibitory control by projections from the prefrontal cortex [108]. However, in some individuals with stress-related disorders, the prefrontal cortex has an excitatory effect on the amygdala [109]. Neuroimaging studies have shown that greater amygdala activity is associated with markers of preclinical atherosclerosis, such as carotid artery intima-media thickness, enhanced blood pressure reactions and inflammatory responses to psychological stressors [98,110,111]. Moreover, higher resting amygdala activity in individuals without known CVD, was associated with increased CVD events during a 4-year follow-up, thought to be mediated by increased bone-marrow activity and release of leukocytes, leading to arterial inflammation [112,113].

***The hypothalamus*** integrates autonomic afferent signals from the brainstem with influences from cortical and subcortical limbic areas, related to evaluation and memories (hippocampus and amygdala). Through the paraventricular nucleus, the hypothalamus is involved in the autonomic responses to emotional stress and is also the main effector of the endocrine stress response, secreting corticotropin-releasing factor which activates the HPA axis.

Cortical areas such as mPFC and ACC have a critical role in the appropriate cognitive appraisal of negative emotions, individual’s self-perception, the ability to switch between alternative emotional control strategies, as well as coordinating sympathetic and parasympathetic responses to environmental stimuli [93,104,114,115,116,117,118]. Most of these brain structures comprise several functionally different areas. In particular, dorsal and ventral areas of mPFC and ACC exert different sympathetic and parasympathetic responses to stimuli [93].

Close relationships between these cortical and subcortical regions’ activity during mental stress have also been observed in clinical studies. In patients with stable CAD, stress-induced activation of mPFC region is associated with high amygdala, thalamus, and insula activation, resulting in autonomic dysfunction, systemic inflammation, angina, and increased incidence of major CVD events [119,120].

***The insula*** integrates the limbic and autonomic systems. It is extensively involved in the perception of physiological condition inside the body (interoception), [121] incorporating body-to-brain autonomic, neuroendocrine, and inflammatory viscerosensory feedback into subjective emotional states [122]. The insula is also involved in regulating cardiac function [123,124,125]. Some studies have described some degree of functional lateralization in the control of the autonomic system, suggesting that right and ventral posterior insula subregions exert sympathetic control, while left and dorsal anterior regions exert parasympathetic effects [96,123]. Acute emotional stress may modify the control of insula over sympathetic and parasympathetic tone, thus raising plasma catecholamine concentrations and increasing the risk of arrhythmias, such as atrial fibrillation, complex ventricular and supraventricular arrhythmias, sudden death, and direct myocardial damage, including myocytolysis and Takotsubo cardiomyopathy [124,126,127,128,129,130]. Moreover, because the insular cortex is supplied by the middle cerebral artery, especially by perforators of M2 divisions, which is the most common embolic stroke area, it is frequently involved in stroke. This may account for the arrhythmias observed in patients with ischemic stroke and insular involvement. Although arrhythmias are generally considered as the cause of stroke, in some patients they occur several days after the stroke [131,132]. In these patients, the arrhythmias may be the consequence of an insular dysfunction rather than the cause of stroke [133,134,135,136,137].

Connections between the cortico–limbic network and the insula, PAG, nucleus of the tractus solitarius, and ventrolateral medulla constitute the central autonomic network (CAN) involved in processing the emotions induced by external stimuli and modulating a coordinated response through the autonomic nervous system (ANS) and neuroendocrine system [138,139]. CAN is also involved in a negative feedback circuit in which the prefrontal cortex exerts tonic inhibitory control over limbic responses to stress [92].

## 5. Stress Effector Systems

Exposure to emotionally stressful stimuli is translated into physiological reactions, called “fight or flight response”, through the autonomic nervous system (ANS) and HPA axis. Cortico–limbic outputs converge on subcortical areas, mainly the hypothalamus and on the nucleus of the tractus solitarius (NTS) of the medulla, which provide direct synaptic connections to sympathetic and parasympathetic preganglionic neurons. NTS receives afferents from the hypothalamus as well as from peripheral baroreceptors and chemoreceptors, modulating the activity of sympathetic and parasympathetic innervations. Finally, autonomic preganglionic efferent fibers synapse with postganglionic fibers within the heart and vasculature. In conditions of stress, amygdala stimulates the hypothalamus and activate the adrenal gland to secrete adrenaline. This in turn stimulates the production of noradrenaline by the locus caeruleus (the principal source of noradrenaline-producing neurons) which has direct connections with amygdala and hypothalamus, thus modulating the stress response [92].

The autonomic nervous system directly innervates the myocardium, coronary arterioles, and conduction system. The heart is under dual autonomic control, sympathetic, and parasympathetic nerves which modulate arterial tone, directly acting on vascular muscle cells and stimulating nitric oxide (NO) release from the endothelium [140]. Sympathetic nerves stimulate β1-adrenergic receptors, resulting in positive chronotropic and inotropic effects, and α1-adrenergic receptors, resulting in vasoconstriction and increase in blood pressure. The density of sympathetic innervation in the coronary arteries is inversely proportional to the vessel caliber. Catecholamines regulate the tone of the pre-arterioles (<500 µm), the epicardial component of coronary microcirculation, responsible for 25% of the total coronary vascular resistance [141]. They respond to wall shear stress through an endothelium-dependent mechanism to maintain pressure of arterioles within a narrow range despite wide changes in coronary perfusion pressure. In normal conditions, an increase in flow rate induces vasodilatation, whereas a reduction results in vasoconstriction [142]. Within seconds after exposure to a stressor, there is an immediate increase in sympathetic nervous system activity, with release of catecholamines, adrenaline, and noradrenaline, into the circulation, and a reduction in parasympathetic cardiac control.

The sympathetic system has also a humoral component, providing most of the circulating adrenaline and some of the noradrenaline from the adrenal medulla, which control myocardial blood flow and microvascular resistance. Adrenal medulla activation is crucial for maintaining high levels of stress arousal for prolonged periods. The sympathetic system also activates the renin–angiotensin–aldosterone system (RAAS). Renin catalyses the formation of angiotensin II which is a powerful vasoconstrictor and also stimulates the release of aldosterone by the adrenal cortex. Aldosterone increases the reabsorption of sodium and water in the kidney, thus modulates blood pressure and influences the process of atherosclerosis by increasing vascular inflammation [143,144]. The parasympathetic efferent nerves exit the medulla forming the vagus nerve (cranial nerve X) and synapse with postganglionic fibers within the heart, vessels, and conduction system. The vagal innervation releases acetylcholine which reduces the intrinsic rate of the sinus node.

At rest, the heart is under tonic inhibitory control by the parasympathetic system, in dynamic balance with the sympathetic system, so that the activity of the two branches can be rapidly modulated in response to changing demands. Exposure to emotional stress reduces the inhibition of prefrontal cortex over limbic control of autonomic responses to stress, resulting in autonomic imbalance with a shift towards increased sympathetic tone and withdrawal of parasympathetic tone. This is expressed by reduced heart rate variability, although the relevance of this measurement as a marker of future CVD is not yet established [145,146]. The vagus nerve also has afferent sensory pathways which transmit signals from baroceptors in the heart and carotid bodies to the NTS and hypothalamus, providing feedback from the CV system thus modulating Its control by the CNS [147].

HPA axis activation is slower than the autonomic nervous system, occurring within minutes after exposure to a stressor. The hypothalamus secretes corticotropin-releasing hormone which controls the release of adrenocorticotropic hormone from the anterior pituitary. This in turn stimulates the adrenal glands to secrete cortisol, promoting the mobilization of stored energy. In normal conditions, cortisol is secreted in pulsatile fashion. During acute stress, the frequency and the amplitude of the hormone release is greatly increased [148]. The HPA axis is subject to feedback inhibition by cortisol, mediated by limbic structures, thus limiting the duration of the total tissue exposure to the effects of cortisol [149].

## 6. Hemodynamic Response to Mental Stress

Acute mental stress is associated with an increase in heart rate and blood pressure, as well as peripheral microvascular constriction, due to sympathetic nervous system activation [150]. The hemodynamic response to acute mental stress is different from that to acute physical exercise (Table 1). During physical exercise, increased sympathetic activity and reduced vagal tone result in increased blood pressure, heart rate, stroke volume, and cardiac output, leading to increased myocardial oxygen demand, expressed by rate-pressure product. Myocardial blood flow increases through dilatation of both epicardial and microvascular resistance coronaries to match myocardial oxygen demand [151]. Myocardial ischemia induced by physical exercise is largely caused by increased workload exceeding coronary blood supply. During exercise, the systemic vascular resistance (SVR) fall due to sympathetic β_2_-adrenergic stimulation and release of locally acting vasodilators, thus reducing left ventricular afterload and allowing increased oxygen delivery to skeletal muscles.

Compared to physical exercise, the response to acute mental stress induces a lower increase in rate-pressure product because heart rate increases are usually less than with exercise. Hence, the rise in cardiac output and myocardial oxygen demand are less. Moreover, in contrast to physical exercise, a mentally stressful task induces activation of endothelial cells and smooth muscle cells of the tunica media by sympathetic α-adrenergic stimulation, resulting in systemic vasoconstriction and elevated blood pressure [152,153,154]. The increase in SVR is a markedly different hemodynamic response compared to physical demand. Mental stress does not need to prepare the body for action, the fight or flight response. Hence, there is no increase in blood flow to big muscle, as occurs with physical demands, but rather a decrease induced by peripheral vasoconstriction. Thus, mental stress results in a hemodynamic mismatch with an increase in left ventricular afterload and myocardial oxygen demand, in the face of augmented SVR. These changes are more marked in patients who develop MSIMI than in those without ischemia, representing a maladaptive response [155].

Due to the rise in SVR, MSIMI occurs at a lower rate-pressure product and oxygen demand than exercise-induced ischemia [36,43,58]. High SVR increases afterload and myocardial wall tension, thus increasing myocardial work and oxygen demand, compromising endocardial perfusion and promoting the development of ischemia. These hemodynamic changes account for the strong inverse correlation between changes in SVR and left ventricular (LV) systolic function or regional wall motion assessed by echocardiography, in response to mental stress [156,157]. Furthermore, with mental stress in patients with CAD, there is only poor concordance between myocardial perfusion and LV wall motion [158]. These findings indicate that reduced LV function during mental stress may not necessarily be a consequence of inadequate blood supply due to obstructive epicardial CAD, leading to myocardial ischemia. Instead, they may reflect changes in loading conditions on the heart due to the effect of stress on peripheral vascular resistance.

## 7. Vascular Response to Mental Stress

Epicardial arteries have a conductance function, exerting only minimal resistance to flow, with diameter regulated by endothelial function and wall shear stress (WSS). Microcirculation is responsible for most of the coronary resistance and for blood distribution according to the myocardial oxygen requirements [140]. Pre-arterioles and arterioles are the epicardial and intramyocardial components of the microcirculation, respectively. Their function matches myocardial blood supply with oxygen demand, and is mainly regulated by endothelium-dependent flow dilatation, mediated by WSS. Sympathetic and parasympathetic nerves innervate coronary resistance vessels and modulate their tone through direct effects on vascular smooth muscle cells as well as by stimulating the release of nitric oxide (NO) from the endothelium.

Mental stress may induce myocardial perfusion abnormalities. However, the degree of perfusion impairment is different in healthy individuals and in CAD patients. As the rate-pressure product at which MSIMI occurs is lower compared with exercise-induced ischemia, there must also be inadequate myocardial blood supply, in addition to increased myocardial oxygen demand, for MSIMI development. While the severity of myocardial perfusion abnormalities with exercise or pharmacologic test is directly related to the fixed effects of coronary stenosis, the risk of MSIMI is due to impaired flow secondary to paradoxical vasoconstriction [30]. Quantitative coronary angiography have shown that in patients with CAD the vasomotor response to mental stress correlates with the extent of atherosclerosis. A high degree of vasoconstriction occurs at points of stenosis, a mild degree occurs in irregular coronary segments, whereas normal segments do not change or dilate. Changes in coronary artery diameter in response to mental stress have been shown to correlate with the local response to the infusion of acetylcholine, an agent used to assess endothelial vasodilator function [159,160,161]. Normally, the sympathetic control of vasomotor tone is negligible and depends on the balance between β-adrenergic mediated arterial dilatation, which is prevalent, and α-adrenergic mediated vasoconstriction that has only a limited effect. In individuals with normal coronary arteries, the α-adrenergic-mediated vasoconstriction in response to high sympathetic stimulation induced by mental stress is counterbalanced by endothelium-mediated vasodilatation. However, in patients with atherosclerotic endothelial dysfunction, the α-adrenergic-mediated vasoconstriction is unrestrained and becomes more intense, reduces blood flow, and may result in myocardial ischemia [154,159]. Imaging studies have shown that myocardial blood flow responses to mental stress in CAD patients is blunted in regions without significant epicardial stenosis, suggesting microvascular dysfunction [26,162,163,164]. Microvascular endothelial dysfunction accounts for approximately two-thirds of myocardial ischemia without significant coronary artery stenosis (<50% diameter stenosis), termed “ischemia with non-obstructive coronary artery disease” (INOCA), which includes microvascular angina, epicardial vasospastic angina, and “myocardial infarction with non-obstructive coronary artery disease” (MINOCA) (Figure 2) [165,166,167,168,169].

There are close interactions between coronary epicardial atherosclerosis and microvascular endothelial dysfunction [170]. Diffuse epicardial atherosclerosis without focal stenosis, as well as chronic coronary artery stenosis, reduce perfusion pressure along the arterial length, resulting in functional and structural microvascular modifications which impair compensatory microvascular vasodilatation in response to lower perfusion pressure and reduced wall shear stress (WSS). WSS is the tangential force of the mechanical friction exerted by the blood on the vascular endothelial surface. Endothelial cells have specific receptors which sense laminar and disturbed blood flow patterns and translate WSS into biochemical signals that modulate the vascular tone, platelet activity, leukocytes adhesion, and endothelial permeability [171]. Low endothelial WSS induces focal inflammation and atherogenesis through decreased production of NO [172]. Therefore, mental stress may induce coronary microvascular endothelial dysfunction which, in turn, may trigger a reduction in WSS of upstream epicardial arteries, resulting in development of focal epicardial lesions [173,174,175,176,177]. Moreover, repeated and cumulative exposure to stress can lead to microvascular dysfunction resulting in myocardial diastolic dysfunction [157,178].

## 8. Conclusions

Exposure to acute and chronic mental stress is associated with the long-term development of atherosclerosis and triggering of acute cardiac events, independently of traditional risk factors. The pathophysiological basis for these clinical syndromes involves the appraisal of stressors by the central nervous system. The emotions provoked by external stimuli are then processed, compared with previous experience, and modulated by a network of dynamically interconnected areas of the brain. These areas regulate the effect of emotional stress on the autonomic, immune, and neuroendocrine systems. The hemodynamic and vascular responses to mental stress may induce myocardial ischemia in patients with and without significant coronary obstruction. This is often a silent condition and less closely correlated to the severity of CAD than exercise-induced ischemia. In future clinical research, brain imaging studies may reveal individual differences in the response to environmental stress. This may provide prognostic CV information, identifying patients for whom aggressive prevention strategies are needed. In addition, clinical trials on the effects of CV events on the brain response to psychological stress may have prognostic relevance, leading to the need for effective secondary prevention, including behavioral interventions.

## Figures and Tables

**Figure 1 jcm-11-03353-f001:**
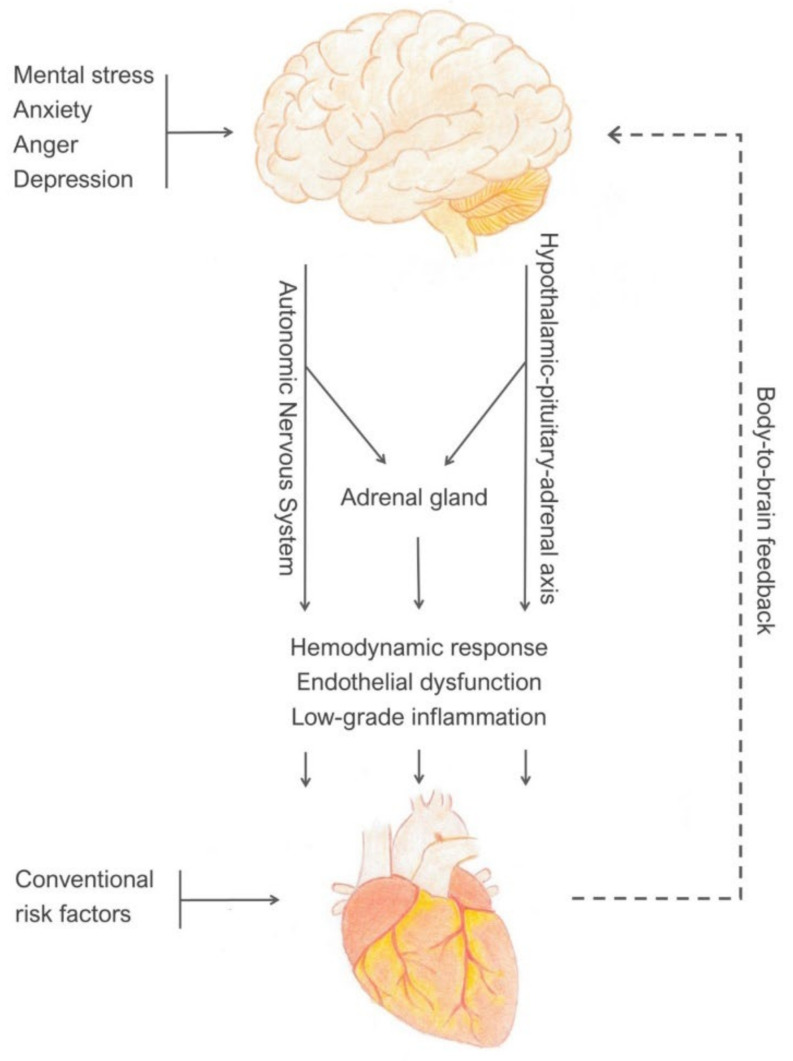
Brain-to-body and body-to-brain control of autonomic, neuroendocrine, and immune responses to mental stress.

**Figure 2 jcm-11-03353-f002:**
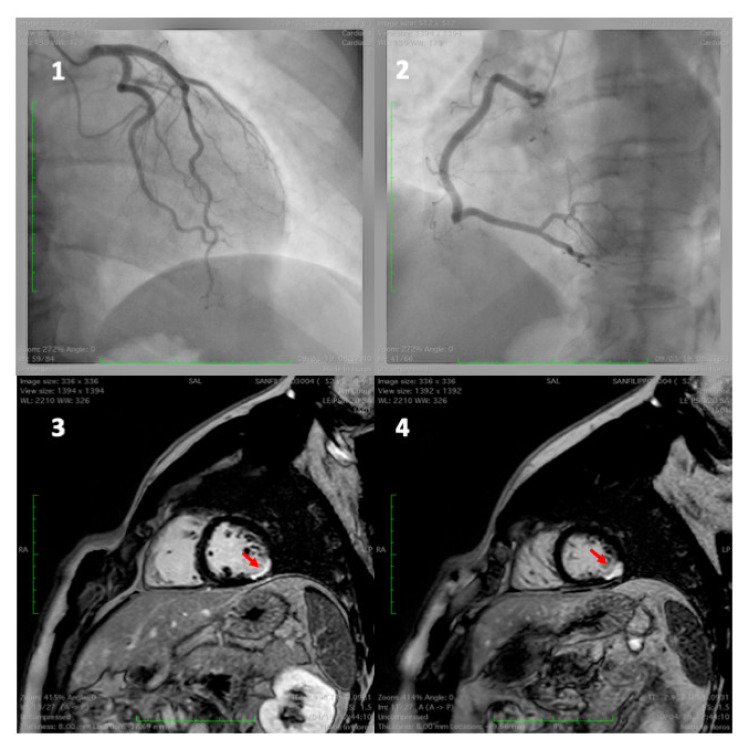
A 49-year-old man admitted for inferior ST elevation myocardial infarction. Coronary angiography showed no coronary lesions on left and right coronary artery (**frames 1** and **2**). The MRI, one day later, detected ischemic lesions (delayed enhancement) of sub-endocardial (**frame 3**) and transmural (**frame 4**) inferior–lateral wall.

**Table 1 jcm-11-03353-t001:** Hemodynamic responses to physical exercise and mental stress.

	Physical Exercise	Mental Stress
Cardiac output	↑↑	↑
Rate-pressure product	↑↑	↑
Myocardial oxygen demand	↑↑	↑
Systemic vascular resistance	↓	↑
LV afterload	↓	↑

Physical exercise is associated with greater increase in cardiac output and rate-pressure product, resulting in greater increase in myocardial oxygen demand. Systemic vascular resistance and left ventricular (LV) afterload fall during exercise while increase in response to mental stress.

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
