# Peer review of "Mental Stress and Cardiovascular Health—Part I"

_jcm, 2022, doi:10.3390/jcm11123353_

Round 1

Reviewer 1 Report

This is a very interesting review article that clearly demonstrated how the exposure to acute and chronic mental stress affects cardiovascular health.

Comments for minor revision:

  1. The introduction is too long. I would advise on shortening.
  2. If you plan to write a second part on the same topic, I would suggest reducing the present text size and incorporating the 2 parts into a comprehensive article, so as not to diminish the readers’ interest.
  3. Please add a figure/table summarizing the hemodynamic and vascular responses to mental stress.

Author Response

We are thankful for the reviewers’ efforts in reviewing our manuscript and for the valuable suggestions which have strengthened it. We hereby address the raised comments.

Reviewer #1

Reviewer comment 1. The introduction is too long. I would advise on shortening.

Author response. The Introduction has been shortened by the entire last paragraph

Reviewer comment 2. If you plan to write a second part on the same topic, I would suggest reducing the present text size and incorporating the 2 parts into a comprehensive article, so as not to diminish the readers’ interest.

Author response. The second part of this review, illustrating the effects of mental stress on vascular endothelium, inflammation, hypertension, arrhythmia, and cardiomyopathy, has already been submitted to JCM. There should be no overlaps between the two parts.

Reviewer comment 3. Please add a figure/table summarizing the hemodynamic and vascular responses to mental stress.

Author response. A table showing the hemodynamic response to physical exercise and mental stress, has been added (Table 1).

Reviewer 2 Report

This review provides a comprehensive summary of the relationship between mental stress and cardiovascular health, with a focus on the pathophysiological mechanisms on this relationship.

I only have minor comments on this paper.

1. In the introduction, the authors reviewed substantial epidemiological evidence on the association between mental stress and cardiovascular health. However, some individual studies did not find this association (https://pubmed.ncbi.nlm.nih.gov/20194301/; https://pubmed.ncbi.nlm.nih.gov/22975465/). This means the epidemiological findings are not consistent. Could the authors discuss this?

2. Could the authors add some implications for reducing the risk of cardiovascular diseases by the management of mental stress in clinical practice?

3.  Would it be possible to discuss more on the implications for future clinical research in this field?

Author Response

Reviewer #2

Reviewer comment 1. In the introduction, the authors reviewed substantial epidemiological evidence on the association between mental stress and cardiovascular health. However, some individual studies did not find this association (https://pubmed.ncbi.nlm.nih.gov/20194301/; https://pubmed.ncbi.nlm.nih.gov/22975465/). This means the epidemiological findings are not consistent. Could the authors discuss this?

Author response. Thank you for the comment. The possibility that studies on the association between mental stress and cardiovascular disease do not reflect their actual relationship has been added to the Introduction: “Furthermore, when the association between psychological stress and CV disease is based on self-reported stress and health outcomes, there may be a tendency to higher reporting of both stress and symptoms of disease, resulting in a spurious association.”

Reviewer comment 2. Could the authors add some implications for reducing the risk of cardiovascular diseases by the management of mental stress in clinical practice?

Author response. A section on Prevention and Treatment is in Part II.

Reviewer comment 3.  Would it be possible to discuss more on the implications for future clinical research in this field?

Author response. Implications for future clinical research have been added to the Conclusions: “In future clinical research, brain imaging studies may reveal individual differences in the response to environmental stress. This may provide prognostic CV information, identifying patients for whom aggressive prevention strategies are needed. In addition, clinical trials on the effects of CV events on the brain response to psychological stress may have prognostic relevance, leading to the need for effective secondary prevention, including  behavioural interventions. “